# Assessing the Functional Accessibility, Actionability, and Quality of Patient Education Materials from Canadian Cancer Agencies

**Courtney van Ballegooie** [1,2,3,†]**, Devon Heroux** [1,2,*,†] **, Peter Hoang** [4] **and Sarthak Garg** [1,2]

1   Experimental Therapeutics, BC Cancer Research Institute, Vancouver, BC V5Z 1L3, Canada
2   Faculty of Medicine, University of British Columbia, Vancouver, BC V6T 1Z3, Canada
3   Faculty of Chemistry, Simon Fraser University, Burnaby, BC V5A 1S6, Canada
4   Division of Geriatric Medicine, Department of Medicine, University of Toronto, Toronto, BC M5T 2S8, Canada
*   Correspondence: dheroux@bccrc.ca
†   These authors contributed equally to this work.

**Abstract:** Patient education materials (PEM)s were extracted from provincial cancer agencies to determine their organizational health literacy by evaluating the quality, actionability, and functional accessibility (e.g., readability and understandability) of their PEMs. PEMs from 10 provincial agencies were assessed for their grade reading level (GRL), using eight numerical and two graphical readability scales, and underwent a difficult word analysis. The agencies were assessed for PEM quality using two methods (JAMA benchmarks and DISCERN), while actionability and understandability were assessed using the Patient Education Materials Assessment Tool (PEMAT). Seven hundred and eighty-six PEMs were analyzed. The overall average GRL was $9.3 \pm 2.1$, which is above the recommended 7th GRL for health information. The difficult word analysis showed that $15.4\% \pm 5.1\%$ of texts contained complex words, $35.8\% \pm 6.8\%$ of texts contained long words, and $24.2\% \pm 6.6\%$ of texts contained unfamiliar words. Additionally, there was high overlap between the most frequently identified difficult words in the PEMs and the most frequently misunderstood words by cancer patients identified in the literature. Regarding quality indicators, no agency displayed all four indicators according to the JAMA benchmarks and DISCERN scores ranged between 38 (poor) to 66 (excellent). PEMAT scores ranged between 68% to 88% for understandability and 57% to 88% for actionability. PEMs continue to be written at a level above the recommended GRL across all provinces, and there was overall high variability in the quality, understandability, and actionability of PEMs among provincial agencies. This represents an opportunity to optimize materials, thus ensuring understanding by a wider audience and improving health literacy among Canadian cancer patients.

**Keywords:** readability; patient education materials; healthcare quality; health literacy

## 1. Introduction

Organizational health literacy is the degree to which organizations equitably enable individuals to find, understand, and use information and services to inform health-related decisions and actions for themselves and others [1]. As health literacy has a significant impact on patient health and quality of life, the responsibilities of organizations and policy makers have continued to evolve [2–4]. As such, the definition of health literacy was revised in 2020 by the Centers for Disease Control and Prevention (CDC) to emphasize an organization's responsibility for providing accessible and actionable information [1].

While direct interactions with a healthcare team member may be perceived as the most educationally effective method by patients, economic and logistical feasibility may prevent this option, especially considering the recent 2019 coronavirus (COVID) pandemic [5]. Given the restrictions seen during the COVID pandemic, along with increasing accessibility to the internet globally, it comes as no surprise that patients' health-seeking behavior has

adapted to include more digital information options [6]. Thus, it is important for digital information to be easily understood by the target population. The ability to "find, understand, and use information" can be examined using a number of methods, such as assessing the quality ("find"), functional accessibility ("understand"), and actionability ("use") of an organization's online patient education materials (PEM)s. Functional accessibility, defined here as the readability and understandability of the PEM, has been shown to be directly dependent on the grade reading level (GRL) of the patient [7]. It has been shown that the average GRL of Canadians is between the 8th and 9th grade [7]. Recommendations from both the CDC and the National Institutes of Health (NIH) suggest that the GRL of PEMs be at least two GRLs below the national average to better ensure comprehension [8].

Cancer is a complex disease with complicated nomenclature for both its biology and treatment. Therefore, it is imperative that PEMs created for individuals with cancer are carefully crafted to ensure accessibility. Readability has been assessed in a number of healthcare fields, including pediatrics, gerontology, internal medicine, amongst others [9–12]. Healthcare topics within cancer, such as radiotherapy, chemotherapy, and other cancer-specific analyses, including breast cancer, colon cancer, and prostate, have also been investigated [13–16]. Within Canada, only a handful of papers have assessed the health literacy of PEMs related to cancer [15–17]. These studies, however, either covered specific sub-topics within cancer, such as chemotherapy, or only addressed the "understanding" component of health literacy. Due to the limited scope of these studies, there has been no comprehensive examination of all three dimensions of health literacy in PEMs across provincial Canadian cancer agencies. Therefore, the aim of this paper was to examine the PEMs of Canadian cancer agencies in order to determine if there were any deficits in health literacy across the three dimensions and provide recommendations should any gaps in literacy be identified. Herein, we assessed the quality (JAMA benchmarks and DISCERN), functional accessibility (readability formulas, difficult word analysis), and actionability (Patient Education Materials Assessment Tool (PEMAT)) of cancer-related PEMs across Canadian provincial cancer agencies.

## 2. Materials and Methods

### 2.1. Sample Collection

Ten provincial agencies were identified according to the Canadian Association for Provincial Cancer Agencies. From May to June 2022, all internet-based PEMs were extracted from the agencies' websites and are listed in Table 1, along with the number of unique PEMs obtained from each agency. The PEMs included materials describing any cancer-related topic with intended use by patients. If a document was in a Portal Document Format (PDF), it was converted to plain text for further analysis. Text sections of nonmedical information were removed from each of the PEMs before analysis, as well as tables and figures, as previously described [18].

### 2.2. Document Readability Analysis

A readability assessment was then performed, as described by Hoang et al. [18]. Briefly, the GRL of the PEMs was determined using eight numerical scales and two graphical scales. The eight numerical scales were comprised of the Degrees of Reading Power (DRP) and Grade Equivalent (GE) test, Flesch–Kincaid Grade Level (FK), Simple Measure of Gobbledygook Index (SMOG), Coleman–Liau Index (CLI), Gunning Fog Index (GF), New Fog Count (NFC), New Dale–Chall readability formula (NDC), and the Ford, Caylor, Sticht (FORCAST) scale. The two graphical scales included the Raygor Readability Estimate Graph (RREG) and the Fry Readability Graph (FRG). These ten scales are externally validated and are frequently used to evaluate the readability of medical texts. The GRL, can range from a 1st GRL to a university-equivalent GRL. The eight numerical scales organized by provincial cancer agency can be seen in Table 2. Greater granularity of the data using the eight scales for each of the provincial cancer agencies can be seen in Supplementary Figure S1.

The FRG and RREG assessments of each cancer agency can be seen in Supplementary Figures S2 and S3, respectively.

### 2.3. Difficult Word Analysis

The difficult word analysis was implemented as previously described [9]. Analysis included identification of the number and percentage of complex words (three or more syllable words), long words (six or more characters), and unfamiliar words, according to the NDC criteria. All words from the PEMs were also extracted and compared to the NDC word list as well as the New General Service List. Words that appeared in either of the lists, including words with the same base word but different tense, were removed and considered as non-jargon words. All words that appeared in less than three PEMs or had a total frequency below three were excluded from analysis. The top ten most frequently identified words were then extracted and their different tenses were combined. Contextual consideration was taken into account for the names of cities, countries, provinces, and territories. Alternative words were then proposed for these most frequently identified words, either using the Readability Studio Software, the Merriam-Webster Thesaurus, or in consultation with a medical doctor, to identify alternatives that could decrease the difficulty of the word, as seen in Supplementary Table S1. Supplementary Tables S2 and S3 display the data obtained from the difficult word analysis and corresponding statistical analysis.

### 2.4. Quality, Actionability, and Understandability Analysis

A quality, actionability, and understandability analysis was then performed using three well established, validated tools, including DISCERN, JAMA benchmarks, and PEMAT. DISCERN assesses the quality and reliability of consumer health information pertaining to treatment choices by grading 16 questions from 1 (inferior) to 5 (superior) (Supplementary Table S6) [19]. Questions 1–8 assess the reliability of the publication as a trusted source, and questions 9–15 focus on the quality of information regarding treatment. Question 16 is an overall rating of the material. The total DISCERN rating is divided into five categories: excellent (63–75), good (51–62), fair (39–50), poor (27–38), and very poor (<26) [20]. Five randomly selected treatment-related PEMs from each agency were analyzed by two independent reviewers according to the DISCERN criteria. The scores of the two reviewers were then averaged for each of the PEMs and the agency's score was then determined by averaging all of the PEMs analyzed. Next, JAMA benchmarks were used to assess the quality of the accountability of each agency by evaluating the PEM's authorship, references, disclosure, and currency (Supplementary Table S5) [20]. Five PEMs were collected from each agency and evaluated by two independent reviewers for each of the four criteria. Because the scores for the JAMA benchmarks are binary, 1 (meets all criteria) and 0 (does not meet all criteria), the mode was used to determine the score of each agency. Lastly, PEMAT was utilized to assess the understandability and actionability of the PEMs extracted from each agency's website (Supplementary Table S7) [21]. Understandability refers to the ability of readers of varying levels of health literacy to understand key messages, whereas actionability refers to the potential of readers to perform actions based on the information presented. The 26 items were scored as 0 (disagree), 1 (agree), or N/A (not applicable) on 19 questions pertaining to understandability and 7 questions pertaining to actionability by two independent reviewers. Each of the questions were averaged by the two reviewers and the five PEMs assessed for each agency were then averaged to obtain each agency's final score.

### 2.5. Statistics

Table 2 reports the arithmetic mean and standard deviation of each numerical scale. The normality of the datasets were tested using a Shapiro–Wilk test when central limit theorem conditions were not met. Equal variance was tested using a Brown–Forsythe test in order to see if the data would need to be transformed before analysis. Normally distributed data with equal variance then underwent a one-way analysis of variance

(ANOVA). If the data was not normally distributed, a non-parametric test was employed. Multiple comparison tests, such as Tukey's test or Tamhane's T2 test, were utilized to identify the differences between sample means in the ANOVA or non-parametric equivalent (e.g., comparison between agency GRL scores for each numerical readability test). Statistical analyses were performed using Graph Pad Prism 9 software.

## 3. Results

### 3.1. Document Readability Analysis

From the ten provincial cancer agencies, 786 PEMs were assessed (Table 1) and a national average GRL was determined to be $9.3 \pm 2.1$, with a GRL range of 5 to 18. The individual scores and average GRL of the eight readability scales can be seen in Table 2. A significant difference ($p < 0.0001$) was identified in the ANOVA. PEMs from NS were found to be the least difficult to read relative to those of all other provincial agencies. The numerical indicators showed that 87.8% and 52.2% of all PEMs were above the 7th (recommended GRL) and 9th (average Canadian GRL) GRL, respectively. The FRG, as seen in Supplementary Figure S2, ranged from 3rd grade to 17+ (university) reading levels. A total of 94.1% and 80.6% of PEMs exhibited a GRL above seven and nine, respectively, with PEMs from NS having the lowest average GRL and those from NB having the highest average GRL. The RREG (Supplementary Figure S3) ranged from a 3rd GRL to a GRL equivalent to that of a professor level (grade 17). A total of 92.5% and 80.6% of PEMs exhibited grade levels above seven and nine, respectively, with those from NS having the lowest average GRL and those from PE having the highest average GRL.

**Table 1.** Number of patient education materials (PEMs) extracted from each provincial cancer agency.

| Cancer Agency | Province | Number of Unique PEMs |
|---|---|---|
| BC Cancer | British Columbia (BC) | 184 |
| Cancer Control Alberta | Alberta (AB) | 110 |
| Sask Cancer Agency | Saskatchewan (SK) | 59 |
| Cancer Care Manitoba | Manitoba (MB) | 70 |
| Cancer Care Ontario | Ontario (ON) | 64 |
| Ministry of Health and Social Services | Quebec (QC) | 40 |
| Cancer Care Eastern Health | Newfoundland and Labrador (NL) | 68 |
| New Brunswick Cancer Network | New Brunswick (NB) | 17 |
| Health PEI | Prince Edward Island (PE) | 25 |
| Nova Scotia Health Authority | Nova Scotia (NS) | 149 |

### 3.2. Difficult Word Analysis

Overall, the PEMs for all provincial cancer agencies contained $15.4 \pm 5.1\%$ complex words of three or more syllables, $35.8 \pm 6.8\%$ words of six or more characters, and $24.2 \pm 6.6\%$ words that were unfamiliar. Table S1 describes the most frequent difficult words by provincial agency. The majority of these words were found to be medically related terms. Table S2 describes the difficult word analysis data for each of the individual provincial agencies, while Table S3 displays the ANOVA and pairwise comparison of the difficult word analysis, which indicated that PEMS from NS showed a statistical significance as being the least difficult in all three difficult word analyses performed.

### 3.3. Quality, Actionability, and Understandability Analysis

From the quality analysis, it was found that the majority of the agencies (70%) displayed only one JAMA benchmark quality indicator (Figure 1a). Currency was identified as the most commonly reported quality indicator, while no agency displayed the disclosure, authorship, and attribution indicators (Figure 1b). PEMAT was used to assess the understandability and actionability of patient information (Figure 1c). The understandability

score ranged from 68% (SK) to 88% (AB), with an average of 84%, while the actionability varied significantly from 57% (NL) to 88% (AB), with an average of 64%. The DISCERN tool was used to assess the reliability and quality of treatment information with the results by section (e.g., quality and reliability) detailed in Figure 1d. The number of available treatment-related PEMs for the analysis were as follows: 1 (QC and PE), 2 (NL), and 5 or more for the remaining provincial agencies. The average rating for reliability and quality using DISCERN was 3.2 out of 5 in both cases. Figure 1d shows results for the DISCERN total rating, which ranged between 38 and 66 out of 80. Alberta's cancer agency was rated as "excellent", those from BC, NS, and NL were rated as "good", those from SK, MB, ON, QC, and PE were rated as "fair," and NB's provincial agency had a "poor" rating. The mean total DISCERN rating for all ten provincial agencies was 51 out of 80, which was rated as "good". The inter-rater reliability of the two reviewers for PEMAT was 88% and, for DISCERN, 60% (when calculating for an exact score) or 95% (when including a ±1 score difference).

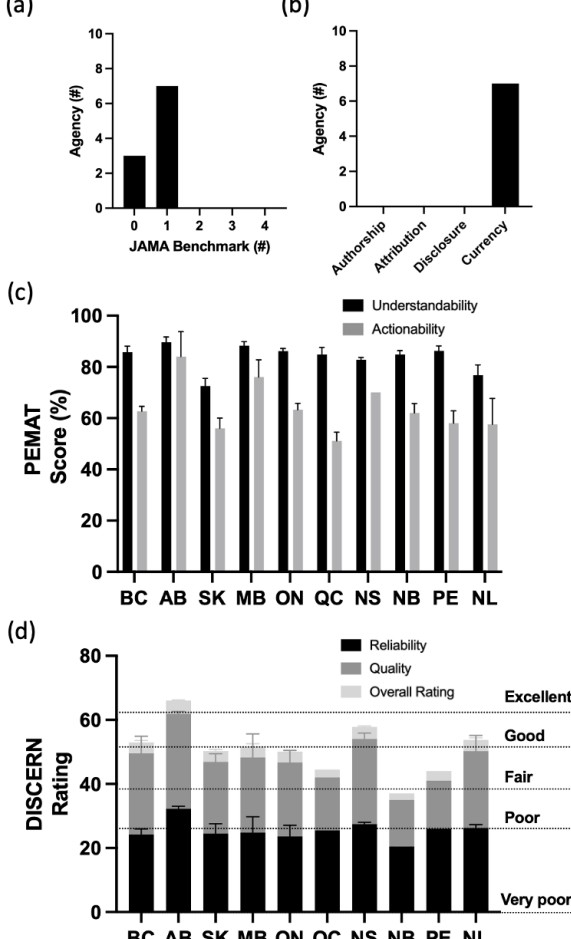

**Figure 1.** The quality, actionability, and understandability analyses using the JAMA benchmarks, DISCERN, and Patient Education Materials Assessment Tool (PEMAT) for PEMs obtained from provincial (Ontario (ON), British Columbia (BC), Nova Scotia (NS), Alberta (AB), Saskatchewan (SK), Québec (QC), Newfoundland and Labrador (NL), Manitoba (MB), Prince Edward Island (PEI), and New Brunswick (NB)) cancer agencies. (**a**) The number of agencies that displayed zero, one, two, three, or all four of the JAMA benchmarks. (**b**) The number of cancer agencies, out of ten, that displayed each of the individual JAMA benchmarks, including authorship, attribution, disclosure, and currency. (**c**) PEMAT scores for the provincial cancer agencies. (**d**) DISCERN ratings for quality and reliability with 80 as a maximum score (see Methods). The # was referring to the categorical numbers of the JAMA benchmarks and the agencies.

**Table 2.** Mean and standard deviation of the grade reading level of online patient education materials for each provincial cancer agency (Ontario (ON), British Columbia (BC), Nova Scotia (SC), Alberta (AB), Saskatchewan (SK), Québec (QC), Newfoundland and Labrador (NL), Manitoba (MB), Prince Edward Island (PEI), and New Brunswick (NB)) using the Coleman–Liau Index (CLI), New Dale–Chall readability formula (NDC), Degrees of Reading Power (DRP) and Grade Equivalent (GE) test, Flesch–Kincaid Grade Level (FK), Gunning Fog Index (GF), New Fog Count (NFC), Simple Measure of Gobbledygook Index (SMOG), and Ford, Caylor, Sticht (FORCAST) scale.

| | CLI | NDC | DRP GE | FK | FORCAST | GF | NFC | SMOG | Average |
|---|---|---|---|---|---|---|---|---|---|
| BC | $10.6 \pm 1.9$ | $9.4 \pm 2.2$ | $9.6 \pm 2.5$ | $8.4 \pm 1.9$ | $10.9 \pm 0.9$ | $9.5 \pm 1.6$ | $4.9 \pm 1.5$ | $10.9 \pm 1.3$ | $9.29 \pm 1.8$ |
| AB | $9.8 \pm 2.6$ | $8.3 \pm 3.1$ | $8.5 \pm 3.7$ | $7.8 \pm 2.5$ | $10.5 \pm 1.2$ | $9.3 \pm 2.2$ | $5.4 \pm 2.0$ | $10.4 \pm 1.8$ | $8.75 \pm 2.5$ |
| SK | $11.7 \pm 2.0$ | $9.5 \pm 2.1$ | $10.5 \pm 3.2$ | $9.8 \pm 2.0$ | $11.0 \pm 0.9$ | $10.3 \pm 1.9$ | $6.3 \pm 2.0$ | $11.8 \pm 1.4$ | $10.1 \pm 2.0$ |
| MB | $10.5 \pm 2.3$ | $9.0 \pm 2.7$ | $9.5 \pm 3.3$ | $8.5 \pm 2.3$ | $10.8 \pm 0.9$ | $9.7 \pm 2.2$ | $5.3 \pm 1.7$ | $10.9 \pm 1.8$ | $9.3 \pm 2.2$ |
| ON | $11.4 \pm 2.4$ | $10.0 \pm 2.7$ | $11.3 \pm 3.3$ | $10.0 \pm 2.8$ | $10.7 \pm 1.1$ | $10.8 \pm 2.6$ | $8.1 \pm 2.7$ | $12.0 \pm 2.2$ | $10.5 \pm 2.5$ |
| QC | $11.0 \pm 1.6$ | $9.6 \pm 2.1$ | $10.1 \pm 2.2$ | $8.9 \pm 1.6$ | $11.3 \pm 0.8$ | $10.3 \pm 1.6$ | $5.6 \pm 1.4$ | $11.1 \pm 1.2$ | $9.7 \pm 1.6$ |
| NS | $9.0 \pm 1.8$ | $7.4 \pm 2.0$ | $7.4 \pm 2.2$ | $7.2 \pm 1.6$ | $10.3 \pm 0.8$ | $8.3 \pm 1.6$ | $4.6 \pm 1.6$ | $9.8 \pm 1.4$ | $8.0 \pm 1.7$ |
| NB | $11.0 \pm 1.7$ | $9.6 \pm 1.7$ | $10.6 \pm 2.1$ | $9.7 \pm 1.5$ | $11.1 \pm 0.8$ | $10.8 \pm 1.4$ | $7.0 \pm 1.5$ | $12.0 \pm 1.2$ | $10.2 \pm 1.5$ |
| PE | $11.6 \pm 1.6$ | $9.9 \pm 1.6$ | $11.4 \pm 2.2$ | $10.3 \pm 1.1$ | $11.1 \pm 0.9$ | $10.7 \pm 1.5$ | $6.5 \pm 1.9$ | $12.3 \pm 1.0$ | $10.5 \pm 1.3$ |
| NL | $11.6 \pm 2.6$ | $9.6 \pm 2.3$ | $11.1 \pm 3.7$ | $9.7 \pm 2.5$ | $11.3 \pm 0.9$ | $10.4 \pm 2.4$ | $5.5 \pm 0.4$ | $11.6 \pm 2.1$ | $10.1 \pm 2.4$ |

## 4. Discussion

### 4.1. Implications

PEMs that are easy to find, understand, and use are critical in order to empower the health literacy of cancer patients. Organizations, such as provincial cancer agencies, have a responsibility to equitably facilitate patients' health-related decision-making capabilities and actions by providing high quality, easily accessible, and actionable PEMs. The results found here suggest that only a handful of provincial cancer agencies excelled in all three categories that were assessed; however, all provincial agencies have areas for improvement. Below, we discuss the quality ("find"), functional accessibility ("understand"), and actionability ("use") of the provincial cancer agencies' online PEMs.

The quality of PEMS from provincial agencies was assessed using JAMA benchmarks for all available PEMs and DISCERN for treatment-related PEMs. While provincial cancer agencies should be regarded as reliable and accessible sources of information, the results suggested that there is a lack of quality with respect to accountability (e.g., ownership of the PEM, including authorship, attribution, disclosure, and currency) across all provinces. Although we were unable to compare the results with those of other health-related agencies across Canada, as this is the first analysis of its kind, the assessment of quality as it pertains to cancer-related information found through web searches has been published. In these studies, it was found that the majority of organizations identified displayed at least two JAMA benchmarks [22–26]; therefore, the current study's findings suggest that provincial cancer agencies have poorer quality on average than those found on the web when assessing authorship, attribution, disclosure, and currency. Parameters that overlapped between JAMA benchmarks and DISCERN (e.g., attribution and currency with questions 4 (source) and 5 (date produced)) were found to be consistent, suggesting that no matter the type of PEM, source and currency were not reliably provided. In fact, questions 4 and 5, as well as questions 1 and 12, had the lowest DISCERN scores across the provincial agencies, with mean scores of 1.5, 2.9, 2.7, and 1.1, respectively. One exception to this, AB, had a perfect rating for question 1 by having explicit aims located at the beginning of their treatment-related PEMs. Finally, the majority of PEMs did not describe what would happen without treatment (question 12) and there was little discussion about alternative treatment options relative to the one presented (question 14).

Our data showed that treatment alternatives and consequences of discontinuation were not addressed in cancer-related PEMs and this omission may result in greater treatment discontinuation rates and, in some cases, potential overtreatment of patients that may choose palliation rather than curative therapy. One of the most prominent examples of

low adherence rates in cancer treatment is adjuvant hormonal therapy in estrogen receptor positive breast cancer. Long term (≥3 years) discontinuation rates can reach over 20% in patient populations [27–29], despite the well-documented correlation between patient survival and long-term adherence. Many speculations have been made about the drivers of low adherence, such as patients weighing their concerns over the necessity of treatment, medication costs, forgetfulness, etc.; however, many new clinical trials address low adherence through increased health literacy via methods such as teaching survivor strategies and self-management, integration of technology for simplified communication and reminders, and integration of interactive interventions [28,29]. A patient's understanding of medication adherence, treatment options, and alternatives is fundamental to their health literacy and ability to make informed decisions [30]. While the underlying reasons that drive non-adherence are complex [27–29], improving health literacy as a multi-component strategy to improve outcomes in clinical trials has been established [28,29].

The functional accessibility of the PEMs from provincial agencies was assessed using a readability analysis and PEMAT. Eight numerical scales and two graphical scales were selected to ensure that no single scale/measure skewed the results to only one variable (e.g., such as FORCAST, which utilizes only a weighted syllabic count; Table S4). Additionally, a difficult word analysis was performed in order to provide recommendations tailored to provincial agencies. In the readability analysis, it was identified that only two provinces (AB and NS) provided PEMs with an average GRL below the national average (9th grade). Nevertheless, all provincial agencies were still above the recommended 7th GRL suggested for health materials to enable comprehension. These findings are in line with other Canadian cancer readability analyses assessing pharmaceutical and COVID-related PEMs, in addition to cancer-related PEMs from the American Medical Association and those identified on the web [14–16,25,31]. As readability does not only pertain to written text, PEMAT was included in the analysis to account for visual aids, layout, and style considerations. While most PEMs scored well for layout and style (e.g., using large font size, chunking text, logical flow, etc.), there was a notable lack of visual aids, with only 20/50 PEMs having "images that could make content more easily understood". Visual aids are particularly important in health education materials for people with low-literacy levels [32,33], and providing images can improve the effectiveness of the material and broaden the target audience. In addition to including visual aids, a difficult word analysis identified key words that could be changed in order to better enable comprehension (Supplementary Table S1). Difficult words that were identified frequently in PEMs across provincial agencies (e.g., in >50% of provinces) included: (1) cervical (cancer), (2) colorectal (cancer), (3) abnormal, (4) diagnose, (5) palliative, (6) radiation, (7) chemotherapy, and (8) medication. While few studies have been performed to identify cancer patients' understanding of individual words they may encounter in PEMs, one study by Pentz et al. assessed patients' understanding of chemotherapy terminology [34]. In this study, they found that words such as palliative, cancer, radiation, and chemotherapy were misunderstood 96%, 74%, 70%, and 58% of the time, respectively. As these words are both regularly misunderstood and are frequently present in PEMs, it is imperative that definitions of these terms be included often and that lay terms be utilized whenever possible. Through the combination of patient comprehension studies and difficult word analyses, cancer agencies can identify difficult words that appear frequently and amend PEMs to enable increased comprehension through greater use of lay terms, an overall decrease in sentence complexity (e.g., by average syllabic count and/or word length), and frequent use of definitions.

Actionability refers to the practical implementation of information for the reader and has gained increasing interest in recent years in patient education [21]. Having sufficient knowledge, skills, and confidence to adequately manage a chronic disease indicates high levels of patient action [35], which has been associated with greater screening rates and survival in colon and breast cancers [36,37]. PEMAT considers material actionable when diverse users can identify their next steps based on the information presented [21]. As a threshold of 70% is considered actionable, only two provinces (AB and MB) met this

standard, with an overall average of 64% denoting poorly actionable material. This was comparable to a recent study of cancer centers in Ontario having a mean actionability of 68% for systemic therapy-related PEMs [37]. The omission of visual aids contributed to low scores in the actionability section, with only 13/50 PEMs having "tangible tools for actions" and 7/50 having "visual aids to act on instructions." Combining visual aids with a verbal description requires less cognitive load for the reader and provides a deeper understanding [38]. This method of including visual aids, such as pictograph-based approaches, has been effective in colonoscopy and radiotherapy preparation [38,39], decision making [40], and risk management [41]. Finally, 0/50 PEMs had a summary section at the end of the material, which would be an excellent opportunity to present the most important points of the material for those with limited literacy.

*4.2. Limitations*

There were a number of limitations to the study described here both in terms of the analyses used as well as the greater context of cancer-related PEMs within Canada. PEMs were only extracted from provincial cancer agency websites, which does not consider the information patients could obtain in person or through other online organizations, such as the Canadian Cancer Society. Furthermore, this study obtained PEMs only written in the English language, which can limit the generalizability of the results. Additionally, the results from this study cannot be generalized to other clinical contexts, such as other medical fields within Canada nor in medical settings outside of Canada. Other parameters that may impact comprehension, such as health and cultural experiences, as well as the specific goals patients may have when reading PEMs, were not taken into consideration [42]. With respect to the analyses, PEMs often contain sections of non-narrative text, such as bullet points, which do not always contain punctuation at the end of the sentence. To account for this, some publications alter the text to consistently contain or exclude punctuation for non-narrative text sections. These changes will result in low and high GRLs, respectively. Authors who have performed this alteration, however, have not found significant changes in their GRLs between the "low" and "high" scenarios [14,18]. Therefore, current strategies may not best capture the benefits these non-narrative text sections provide with respect to readability. To overcome this limitation, this study employed PEMAT in order to better consider the organization, layout, and design of the PEMs and the impact their non-narrative text may have on understandability. In addition to the non-narrative text, other limitations can also be identified in the readability analysis. For example, while words such as oncology are likely to be understood by many patients, some readability scores use a component of syllabic count in their calculation. This in turn would lead to an overestimation of the GRL. On the other hand, medical jargon containing few letters and syllables may evade detection on readability tests that rely on the number of letters and syllables. The difficulty word analysis also identified words as complex based on their syllabic count, which would face similar limitations to those described above. Future studies should pursue the inclusion of direct patient interviews to better understand which terms patients may or may not understand in the context of PEMs produced by Canadian provincial cancer agencies.

**5. Conclusions**

Organizational health literacy was investigated across provincial cancer agencies by analyzing online PEMs with regards to their quality, functional accessibility, and actionability. Overall, this study demonstrated serious deficits in organizational health literacy across provincial cancer agencies. These deficits present opportunities for provincial cancer agencies to improve their organizational health literacy in order to promote the health literacy of individual patients, thus better informing their health-related decisions and actions. Specific actions that provincial cancer agencies can undertake to improve the health literacy of cancer patients include incorporation of the following in all treatment-related PEMs: (1) visual diagrams, (2) definitions or replacement of difficult words, (3) source

data and currency, (4) introductory document aims, (5) a summary/key takeaways, and (6) information surrounding treatment options and palliation.

**Supplementary Materials:** The following supporting information can be downloaded at: https://www.mdpi.com/article/10.3390/curroncol30020110/s1, Figure S1: Box Plot of Mean Grad Level of Online Patient Education Materials Found on Each Provincial Cancer Agency using 8 Numerical Scales; Table S1: Difficult Words with Alternative Word Recommendations; Figure S2: Fry Readability Graph Assesment of Online Patient Education Materials; Figure S3: Raygor Readability Estimate Graph of Online Patient Education Materials; Table S3: Difficult Words Analysis Statistics; Table S4: Readability Formulas; Table S5. JAMA Benchmark Criteria; Table S6 DISCERN Instrument Criteria, Table S7: PEMAT Criteria

**Author Contributions:** Conceptualization, C.v.B.; methodology, C.v.B.; software, C.v.B.; formal analysis, C.v.B.; investigation, C.v.B., P.H., D.H. and S.G.; resources, C.v.B.; data curation, C.v.B., P.H., D.H. and S.G.; writing—original draft preparation, C.v.B., P.H., D.H. and S.G.; writing—review and editing, C.v.B. and D.H.; visualization, C.v.B.; supervision C.v.B.; funding acquisition, C.v.B. All authors have read and agreed to the published version of the manuscript.

**Funding:** C.v.B. was supported by a Canadian Graduate Scholarships Doctoral Program, Canadian Institutes of Health Research (CGS-D CIHR; Grant No. 21R04868), an NMIN salary grant (Grant No. 10901), and a University of British Columbia Four Year Doctoral Graduate Fellowship (Grant No. 6569).

**Institutional Review Board Statement:** This study qualifies for exemption of ethical approval due to the use of non-human subject material (patient education materials).

**Informed Consent Statement:** Written informed consent is not applicable due to the use of non-human test subjects (patient education materials).

**Data Availability Statement:** Data can be made available upon request.

**Conflicts of Interest:** The authors declare no conflict of interest. The funders had no role in the design of the study; in the collection, analyses, or interpretation of data; in the writing of the manuscript; or in the decision to publish the results.

## Abbreviations

| | |
|---|---|
| AB | Alberta |
| ANOVA | Analysis of Variance |
| BC | British Columbia |
| CDC | Centers for Disease Control and Prevention |
| CLI | Coleman–Liau Index |
| COVID | Coronavirus |
| DRP | Degrees of Reading Power |
| FK | Flesch–Kincaid Grade Level |
| FORCAST | Ford, Caylor, Sticht |
| FRG | Fry Readability Graph |
| GR | Grade Equivalent |
| GRL | Grade reading level |
| GF | Gunning Fog Index |
| MB | Manitoba |
| NIH | National Institutes of Health |
| NB | New Brunswick |
| NDC | New Dale–Chall |
| NFC | New Fog Count |
| NL | Newfoundland and Labrador |
| NS | Nova Scotia |
| ON | Ontario |

| PEM | Patient education materials |
| PEMAT | Patient Education Materials Assessment Tool |
| PEI | Prince Edward Island |
| PDF | Portal Document Format |
| QC | Québec |
| SK | Saskatchewan |
| SMOG | Simple Measure of Gobbledygook Index |
| RREG | Raygor Readability Estimate Graph |

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
