# Peer review of "Assessing the Functional Accessibility, Actionability, and Quality of Patient Education Materials from Canadian Cancer Agencies"

_curroncol, doi:10.3390/curroncol30020110_

Round 1
Reviewer 1 Report
Pretty interesting work. The idea of evaluating PEMs is excellent and different tools are used to assess them. The thing I was most curious about while reading the manuscript was this. It was mentioned that 2 independent reviewers evaluated PEMs in terms of quality, content, etc. Did you also statistically examine the reliability and/or agreement between the interraters? As it is known, interrater reliability is crucial because it represents the extent to which the data collected in the study are correct representations of the variables measured
Author Response
The authors would like to thank the reviewer for their comment. Interrelator reliability was calculated for PEMAT (88%) on a binary scale, and DISCERN (95% for ±1 score) on a five-point scale. This information has been added to Section 3.3.
Reviewer 2 Report
For the readability analysis and difficult word analysis, it would be helpful to have a brief description of how each were performed even if the specifics of the analysis are documented elsewhere. How many reviewers examined the text? If more than one, how were conflicts in ratings handled?
Why were so many reading level rating scales chosen (other than that is what Hoang et al did)? It is not clear in the discussion what each measure is actually assessing and how they may be similar or different. From the figure (which is very difficult to read), the take away really appears to be that some of the measures have more stringent criteria than others when assessing reading grade level, not that there is any difference between the provinces or organizations that were reviewed.
In the statistical analysis section, it is not clear which variables are being compared. Is it a comparison between each outcome measure across each organization?
Which nonparametric test was used for data that was not normally distributed?
Figure 1 is nearly impossible to read because it is so cluttered. Why is the analysis broken up by province rather than producing overall results? The figure would be much easier to read if it was collapsed across all provinces and the research question of the study does not seem to indicate that a comparison across provinces is the goal of the study.
It would be helpful to have a description of each of the measures and what is included in each measure in a table so the reader can easily see what each is assessing, where there is overlap, and where they diverge.
Quantitative data should be presented in tables as it will be much easier for the reader to assess than trying to parse results from the narrative.
In the discussion, there is talk about treatment alternatives and consequences to discontinuation not being discussed in the cancer PEMs, but that issue is never brought up in the introduction, methods, or results so it is unclear how that is relevant to the paper.
Author Response
- For the readability analysis and difficult word analysis, it would be helpful to have a brief description of how each were performed even if the specifics of the analysis are documented elsewhere. How many reviewers examined the text? If more than one, how were conflicts in ratings handled?
- The authors would like to thank the reviewer for their comment. A supplementary table has been added to provide the specific analyses for each of the readability formulas. Because these formulas were performed using readability software (readability studio) there were no conflicts in the scores.
- The authors would like to thank the reviewer for their comment. A supplementary table has been added to provide the specific analyses for each of the readability formulas. Because these formulas were performed using readability software (readability studio) there were no conflicts in the scores.
- Why were so many reading level rating scales chosen (other than that is what Hoang et al did)? It is not clear in the discussion what each measure is actually assessing and how they may be similar or different. From the figure (which is very difficult to read), the take away really appears to be that some of the measures have more stringent criteria than others when assessing reading grade level, not that there is any difference between the provinces or organizations that were reviewed.
- The takeaway from figure one is that no matter the readability test utilized (all of which are validated and used to evaluate medical text, as described in the methods) the majority of them have an average grade reading level above the recommended 7th grade reading level (and, most of them even above the national 9th grade reading level). An additional descriptor line has been added to highlight the use of many readability tests (e.g. to not skew the results to only one or few parameters, such as sentence length, etc.). The new line reads as follows: “Eight numerical scales and two graphical scales were selected to ensure that no single scale/measure skewed the results to only one variable (e.g. such as FORCAST which utilizes only a weighted syllabic count; Table S4).” Additionally, Table S4 has been added to the supplementary with a more detailed section describing the make-up of the readability scales.
- The takeaway from figure one is that no matter the readability test utilized (all of which are validated and used to evaluate medical text, as described in the methods) the majority of them have an average grade reading level above the recommended 7th grade reading level (and, most of them even above the national 9th grade reading level). An additional descriptor line has been added to highlight the use of many readability tests (e.g. to not skew the results to only one or few parameters, such as sentence length, etc.). The new line reads as follows: “Eight numerical scales and two graphical scales were selected to ensure that no single scale/measure skewed the results to only one variable (e.g. such as FORCAST which utilizes only a weighted syllabic count; Table S4).” Additionally, Table S4 has been added to the supplementary with a more detailed section describing the make-up of the readability scales.
- In the statistical analysis section, it is not clear which variables are being compared. Is it a comparison between each outcome measure across each organization?
- An additional line has been added to the text to better clarify this. It reads as “Multiple comparison’s tests, such as Tukey’s tests or Tamhane’s T2 test, were utilized to identify differences between sample means in the ANOVA, or non-parametric equivalent, analysis (e.g. comparing between provinces’ GRL score for each numerical readability test).”
- Which nonparametric test was used for data that was not normally distributed?
- The authors would like to thank the reviewer for their comment. When a non-parametric test was utilized, a Tukey’s test equivalent (Tamhane’s T2 multiple comparisons test) was utilized. This has now been added to the methods section to improve clarity and reads as follows: “Multiple comparison’s tests, such as Tukey’s tests or Tamhane’s T2 test, were utilized to identify differences between sample means in the ANOVA, or non-parametric equivalent, analysis (e.g. comparing between provinces’ GRL score for each numerical readability test)”
- The authors would like to thank the reviewer for their comment. When a non-parametric test was utilized, a Tukey’s test equivalent (Tamhane’s T2 multiple comparisons test) was utilized. This has now been added to the methods section to improve clarity and reads as follows: “Multiple comparison’s tests, such as Tukey’s tests or Tamhane’s T2 test, were utilized to identify differences between sample means in the ANOVA, or non-parametric equivalent, analysis (e.g. comparing between provinces’ GRL score for each numerical readability test)”
- Figure 1 is nearly impossible to read because it is so cluttered. Why is the analysis broken up by province rather than producing overall results? The figure would be much easier to read if it was collapsed across all provinces and the research question of the study does not seem to indicate that a comparison across provinces is the goal of the study.
- To accommodate suggestions 5 and 7 “Quantitative data should be presented in tables as it will be much easier for the reader to assess than trying to parse results from the narrative.”, the figure has now been modified to a table. Additionally, the average of the scores has now been removed from the narrative text and placed in the table as well. A line has been added after table 1 to highlight the national GRL of PEMs and reads as follows “From the ten provincial cancer agencies, 786 PEMs were assessed (Table 1) and a national average GRL was determined to be 9.3 ± 2.1.” After these modifications, all narrative text discussing results can now be found in a table or figure. It should be noted that the goal of the study was to not only determine the national average but also the provincial average as often patient related education material cannot be shared between provinces due to different resources, programs, treatment regimes, etc. found in each.
- To accommodate suggestions 5 and 7 “Quantitative data should be presented in tables as it will be much easier for the reader to assess than trying to parse results from the narrative.”, the figure has now been modified to a table. Additionally, the average of the scores has now been removed from the narrative text and placed in the table as well. A line has been added after table 1 to highlight the national GRL of PEMs and reads as follows “From the ten provincial cancer agencies, 786 PEMs were assessed (Table 1) and a national average GRL was determined to be 9.3 ± 2.1.” After these modifications, all narrative text discussing results can now be found in a table or figure. It should be noted that the goal of the study was to not only determine the national average but also the provincial average as often patient related education material cannot be shared between provinces due to different resources, programs, treatment regimes, etc. found in each.
- It would be helpful to have a description of each of the measures and what is included in each measure in a table so the reader can easily see what each is assessing, where there is overlap, and where they diverge.
- The authors would like to thank the reviewer for their comment. A supplementary table (Table S4) has been added to provide the specific analyses for each of the readability formulas.
- The authors would like to thank the reviewer for their comment. A supplementary table (Table S4) has been added to provide the specific analyses for each of the readability formulas.
- Quantitative data should be presented in tables as it will be much easier for the reader to assess than trying to parse results from the narrative.
- Please see the response to question 5.
- Please see the response to question 5.
- In the discussion, there is talk about treatment alternatives and consequences to discontinuation not being discussed in the cancer PEMs, but that issue is never brought up in the introduction, methods, or results so it is unclear how that is relevant to the paper.
- This is touched on broadly in the introduction when discussing actionability in the context of health literacy (PEMAT). Paragraph 2 of the discussion then hones into two questions of the PEMAT which provinces have performed poorly on (e.g. discussing treatment alternatives and consequences of non-treatment). In order to better link actionability to these questions, all tools which are utilized in the study now have their questions provided in a supplementary table (Tables S5-S7).
Reviewer 3 Report
Very well prepared work. The methodology, selection of sources and statistics do not raise any objections. However, the introductory part, discussion and conclusions seem to be too extensive in relation to the results of the table and statistics. It gets boring at times. The work seems to be a great model for carrying out this type of research in other countries. The results alone are not necessarily interesting for people from Canada.
Author Response
In order to address this comment, the introduction has been shortened and the scope has been narrowed to only include organizational health literacy and not personal health literacy. Additionally, net new data found in the narrative text of the results section has been moved to Table 2 to better engage the audience. Next, the conclusion has had its data, which was already addressed in the results section, removed in order to decrease redundancy and increase engagement. Lastly, to better link the discussion to the results and analysis performed, each of the questions for the tools used has been provided in a supplementary table. In doing so, the authors hope that the content of the discussion is more easily related to the introduction and results (as indicated by another reviewer). Thank you for this comment and please let us know if these were the kinds of revisions you were looking for to address this comment or if there were other tactical edits you would like us to do (specific lines or examples would be very much appreciated if you feel that the above did not properly address the comment).
Reviewer 4 Report
Thank you for the opportunity to review this interesting manuscript on “Assessing the Functional Accessibility, Actionability, and Quality of Patient Education Materials from Canadian Cancer Agencies”, investigating Canadian cancer agencies’ PEMs. I find your paper very interesting!
You can find my comments below:
L95 “The eight numerical scales comprised of…” Higher scores of each scale indicates a higher difficulty to read? What is the range in each scale? Please clarify this in your methodology part. Explain also in this part how you evaluate the two graphical scales.
L158-159 “Multiple comparison’s tests, such as Tukey’s tests, were utilized to identify differences between sample means in the ANOVA analysis.” In non-parametric test, did you also use a correction method in your post hoc analyses?
L173-173 “The numerical indicators showed that 87.8% and 52.2% of PEMs were above the 7th and 9th GRL, respectively.” You should mention why do you use these cut-offs in the methodology part. It was confusing to me to understand these cut-offs until I read the whole paper (I found 9th GRL in the introduction part and 7th GRL in the discussion part).
In L175-179 (Suppl Fig 2 & 3) you should comment the difference in scores between cancer agencies (i.e. which scored the best and the poorest)
L197 “…which indicates that NS showed a statistical significance as being the least difficult in all three difficult word analyses performed.” I don’t clearly understand this. NS represents a “non-significance” correlation, as presented in your table. Please clarify.
L212-214 “Figure 2d shows results for the DISCERN total rating, which ranged between 38 and 66 out of 80. One province (AB) was rated as “excellent”, three provinces (BC, NS, and NL) were rated as “good”, five provinces (SK, MB, ON, QC, and PE) as “fair” and one province had a “poor” rating (NB).” Did you run post hoc analyses in these scores? If yes, could you add some p-values i.e. total p and the most significant or of clinical interest? Could you also do this when interpreting the results from Figure 2c?
L251-252 “In fact, questions 4 and 5, as well as questions 1 and 12, 251 were the lowest scoring DISCERN” Could you add a Supplementary Table including all questions of JAMA, PEMAT and DISCERN because you include some specific questions in you discussion, but these questions are not mentioned anywhere and it is uncomfortable for the reader to search them online.
Author Response
- L95 “The eight numerical scales comprised of…” Higher scores of each scale indicates a higher difficulty to read? What is the range in each scale? Please clarify this in your methodology part. Explain also in this part how you evaluate the two graphical scales.
- The authors would like to thank the reviewer for their comment. An additional line to improve the clarity of the range was added to the methods section and reads as follows “. The GRL, which can range from a 1st GRL to a university equivalent GRL, using the eight numerical scales organized by provincial cancer agency can be seen in Figure 1.” Additionally, a supplementary table has been added so that readers can reference each of the formulas and identify which factors (e.g. sentence length, average number of syllables, etc.) were utilized to determine GRL.
- The authors would like to thank the reviewer for their comment. An additional line to improve the clarity of the range was added to the methods section and reads as follows “. The GRL, which can range from a 1st GRL to a university equivalent GRL, using the eight numerical scales organized by provincial cancer agency can be seen in Figure 1.” Additionally, a supplementary table has been added so that readers can reference each of the formulas and identify which factors (e.g. sentence length, average number of syllables, etc.) were utilized to determine GRL.
- L158-159 “Multiple comparison’s tests, such as Tukey’s tests, were utilized to identify differences between sample means in the ANOVA analysis.” In non-parametric test, did you also use a correction method in your post hoc analyses?
- The authors would like to thank the reviewer for their comment. When a non-parametric test was utilized, a Tukey’s test equivalent (Tamhane’s T2 multiple comparisons test) was utilized. This has now been added to the methods section to improve clarity and reads as follows: “Multiple comparison’s tests, such as Tukey’s tests or Tamhane’s T2 test, were utilized to identify differences between sample means in the ANOVA, or non-parametric equivalent,”
- The authors would like to thank the reviewer for their comment. When a non-parametric test was utilized, a Tukey’s test equivalent (Tamhane’s T2 multiple comparisons test) was utilized. This has now been added to the methods section to improve clarity and reads as follows: “Multiple comparison’s tests, such as Tukey’s tests or Tamhane’s T2 test, were utilized to identify differences between sample means in the ANOVA, or non-parametric equivalent,”
- ‘L173-173 “The numerical indicators showed that 87.8% and 52.2% of PEMs were above the 7th and 9th GRL, respectively.” You should mention why do you use these cut-offs in the methodology part. It was confusing to me to understand these cut-offs until I read the whole paper (I found 9thGRL in the introduction part and 7th GRL in the discussion part).
- The authors would like to thank the reviewer for their comment. L173 has now been modified to better provide clarity surrounding the 7th and 9th GRL benchmarks that were used. The line now reads as follows: “The numerical indicators showed that 87.8% and 52.2% of PEMs were above the 7th (the recommended GRL) and 9th (the average Canadian GRL) GRL, respectively.”
- The authors would like to thank the reviewer for their comment. L173 has now been modified to better provide clarity surrounding the 7th and 9th GRL benchmarks that were used. The line now reads as follows: “The numerical indicators showed that 87.8% and 52.2% of PEMs were above the 7th (the recommended GRL) and 9th (the average Canadian GRL) GRL, respectively.”
- In L175-179 (Suppl Fig 2 & 3) you should comment the difference in scores between cancer agencies (i.e. which scored the best and the poorest)
- The authors would like to thank the reviewers for their comment. L175-179 has now been revised to include the highest and lowest scoring provinces for the two graphical measurements. The lines now read as follows: The FRG, as seen in Supplementary Figure S2, ranges from a 3rd grade to a 17+ (university) reading level. 94.1% and 80.6% of PEMS exhibited a GRL above seven and nine, respectively, with NS having the lowest average GRL and NB the highest average GRL. The RREG (Supplementary Figure S3) ranges from a 3rd GRL to a GRL equivalent to that of a professor level (grade 17). 92.5% and 80.6% of PEMS exhibited a grade level above seven and nine, respectively, with NS having the lowest average GRL and PE the highest average GRL.”
- L197 “…which indicates that NS showed a statistical significance as being the least difficult in all three difficult word analyses performed.” I don’t clearly understand this. NS represents a “non-significance” correlation, as presented in your table. Please clarify.
- NS stands for the province Nova Scotia, as indicated in table 1, rather than non-significance. A list of abbreviations has been added to the paper in order to better clarify this.
- NS stands for the province Nova Scotia, as indicated in table 1, rather than non-significance. A list of abbreviations has been added to the paper in order to better clarify this.
- L212-214 “Figure 2d shows results for the DISCERN total rating, which ranged between 38 and 66 out of 80. One province (AB) was rated as “excellent”, three provinces (BC, NS, and NL) were rated as “good”, five provinces (SK, MB, ON, QC, and PE) as “fair” and one province had a “poor” rating (NB).” Did you run post hoc analyses in these scores? If yes, could you add some p-values i.e. total p and the most significant or of clinical interest? Could you also do this when interpreting the results from Figure 2c?
- An ANOVA was not performed as the data was generated from discrete measurements (e.g. Likert scale data which could only have a discrete number, such as 1, 2, 3, 4, or 5 rather than continuous data)
- An ANOVA was not performed as the data was generated from discrete measurements (e.g. Likert scale data which could only have a discrete number, such as 1, 2, 3, 4, or 5 rather than continuous data)
- L251-252 “In fact, questions 4 and 5, as well as questions 1 and 12, 251 were the lowest scoring DISCERN” Could you add a Supplementary Table including all questions of JAMA, PEMAT and DISCERN because you include some specific questions in you discussion, but these questions are not mentioned anywhere and it is uncomfortable for the reader to search them online.
- The authors would like to thank the reviewer for their comment. Supplementary Tables S5-7 have been added with questions for the three methods, and referenced in Section 2.4.
Round 2
Reviewer 2 Report
Authors have addressed all relevant concerns from prior version.